# Structure of the human volume regulated anion channel

**Jennifer M Kefauver[1,2], Kei Saotome[1,2†], Adrienne E Dubin[1†], Jesper Pallesen[2], Christopher A Cottrell[2], Stuart M Cahalan[1‡], Zhaozhu Qiu[3§#], Gunhee Hong[1], Christopher S Crowley[2,4], Tess Whitwam[1], Wen-Hsin Lee[2], Andrew B Ward[2*], Ardem Patapoutian[1*]**

[1]Department of Neuroscience, Howard Hughes Medical Institute, The Scripps Research Institute, La Jolla, United States; [2]Department of Integrative Structural and Computational Biology, The Scripps Research Institute, La Jolla, United States; [3]Genomics Institute of the Novartis Research Foundation, San Diego, United States; [4]Department of Dermatology, University of California, San Diego, San Diego, United States

**\*For correspondence:**
andrew@scripps.edu (ABW);
ardem@scripps.edu (AP)

[†]These authors contributed equally to this work

**Present address:** [‡]Vertex Pharmaceuticals, San Diego, United States; [§]Department of Physiology, Johns Hopkins University School of Medicine, Baltimore, United States; [#]Solomon H. Snyder Department of Neuroscience, Johns Hopkins University School of Medicine, Baltimore, United States

**Abstract** SWELL1 (LRRC8A) is the only essential subunit of the Volume Regulated Anion Channel (VRAC), which regulates cellular volume homeostasis and is activated by hypotonic solutions. SWELL1, together with four other LRRC8 family members, potentially forms a vastly heterogeneous cohort of VRAC channels with different properties; however, SWELL1 alone is also functional. Here, we report a high-resolution cryo-electron microscopy structure of full-length human homo-hexameric SWELL1. The structure reveals a trimer of dimers assembly with symmetry mismatch between the pore-forming domain and the cytosolic leucine-rich repeat (LRR) domains. Importantly, mutational analysis demonstrates that a charged residue at the narrowest constriction of the homomeric channel is an important pore determinant of heteromeric VRAC. Additionally, a mutation in the flexible N-terminal portion of SWELL1 affects pore properties, suggesting a putative link between intracellular structures and channel regulation. This structure provides a scaffold for further dissecting the heterogeneity and mechanism of activation of VRAC.
DOI: https://doi.org/10.7554/eLife.38461.001

## Introduction

VRAC is a ubiquitously expressed mammalian anion channel implicated in diverse physiological processes including volume regulation, cell proliferation, release of excitatory amino acids, and apoptosis (*Hyzinski-García et al., 2014*; *Nilius et al., 1997*; *Pedersen et al., 2016*). It is suggested to play a role in a variety of human diseases including stroke, diabetes, and cancer (*Hyzinski-García et al., 2014*; *Planells-Cases et al., 2015*; *Zhang et al., 2017*). A causative link has been established between a chromosomal translocation in the *SWELL1* (*LRRC8A*) gene and a human B cell deficiency disease, agammaglobulinemia (*Sawada et al., 2003*).

Previous studies have shown that SWELL1 is required for VRAC activity, and that the presence of other LRRC8 subunits dictates functional characteristics of VRAC, including pore properties (*Qiu et al., 2014*; *Syeda et al., 2016*; *Voss et al., 2014*). While SWELL1 and at least one other LRRC8 subunit are required for canonical whole-cell VRAC currents, purified homomers of SWELL1 reconstituted in lipid bilayers are activated by osmotic stimuli and blocked by VRAC antagonist, DCPIB (*Syeda et al., 2016*). Interestingly, CRISPR-engineered HeLa cells lacking all LRRC8 subunits (*LRRC8$^{-/-}$* HeLa cells) exhibited very small but significant DCPIB-sensitive hypotonicity-induced currents after SWELL1 overexpression (*Figure 1—figure supplement 1*), supporting previous bilayer results. Since the number and composition of functional native oligomeric assemblies remains

**eLife digest** Every cell needs to regulate its internal volume or it will burst. Most of a cell's volume is a watery mixture of salts, proteins and other molecules. A cell can take in more water from its surroundings, diluting this mixture and causing the cell to expand. If a cell starts to take up too much water, it will open channel proteins in its outer membrane called volume regulated anion channels (or VRACs for short). An open VRAC allows negatively charged ions to leave the cell, and in the process causes water to leave the cell too. This relieves the pressure inside the cell, and the cell starts to shrink.

The structure of a VRAC is thought to contain six subunits, and most include at least two different kinds of subunit. Some of the subunits must be a protein called SWELL1 (which is also known as LRRC8A). The other subunits can be any of four similar proteins from the same protein family. Since a VRAC can contain additional subunits drawing from this pool of five proteins, many structures are possible. But it remains unclear exactly how the structure of a VRAC allows it to sense and regulate the volume of a cell. This is partly because scientists do not have enough information about the architecture of this protein to understand how it might work.

Using electron microscopes, Kefauver et al. have now captured detailed images of a VRAC composed entirely of human SWELL1 proteins. The overall structure of VRAC resembles a six-legged jellyfish, with a pore on the cell's exterior passing through a constricted dome followed by three pairs of arms that extend into the cell's interior. Given the observed structure, Kefauver et al. speculate that the arms of the SWELL1 proteins sense salt concentrations within the cell (to tell if its become diluted by an influx of water) and then interact with the rest of the channel. In response to these interactions, the domed part of the VRAC constricts or dilates to help regulate the cell's volume.

Molecular biologists can now use these structural details to further study the fundamentals behind how cells regulate their volume. This model will also improve scientific understanding of how diverse VRAC structures differ in their responses to changes in pressure within cells.
DOI: https://doi.org/10.7554/eLife.38461.002

unknown, we decided to first elucidate the structure of SWELL1 homomers. To produce homomeric SWELL1, human SWELL1-FLAG was recombinantly expressed in *LRRC8(B,C,D,E)$^{-/-}$* HEK293-F suspension cells, then solubilized in 1% decyl maltose neopentyl glycol (DMNG) detergent, followed by purification and exchange into 0.05% digitonin for structure determination by cryo-EM (*Figure 1—figure supplement 2*). Image analysis and reconstruction yielded a ~4 Å resolution map that was used to build a molecular model of SWELL1 (*Figure 1—figure supplements 3–4*, *Supplementary file 1*).

## Results

SWELL1 is organized as a hexameric trimer of dimers with a four-layer domain architecture and an overall jellyfish-like shape (*Figure 1A*). The transmembrane (TM) and extracellular domains (ECDs) surround the central pore axis, and share a previously unappreciated structural homology with the connexin (*Maeda et al., 2009*) and innexin (*Oshima et al., 2016*) gap junction channels (*Figure 1—figure supplement 5A–D*). The ECD is composed of two extracellular loops (ECL1 and ECL2) that are stabilized by three disulfide bonds (*Figure 1B–C* and *Figure 1—figure supplement 5E–F*). ECL1 contains one strand of a small beta-sheet and a helix (ECH) that faces the center of the ECD while ECL2 contains two additional antiparallel beta strands of the beta-sheet that faces the outside of the ECD. Each subunit contains four TM helices (TM1-4). TM1 lies closest to the central pore axis and is tethered to a short N-terminal coil (NTC) that is parallel to the inner leaflet of the membrane. In the cytosol, the intracellular linker domains (ILD) create a tightly packed network of helices connecting the channel pore to the LRR domains. Each ILD is composed of two-four helices from the TM2-TM3 cytoplasmic loop (LH1-4), and five helices from the TM4-LRR linker (LH5-9) (*Figure 1C*). Each protomer terminates in 15–16 LRRs which form a prototypical solenoid LRR fold (*Figure 1B–C*). LRRs from the six protomers dimerize into three pairs, which interact to form a Celtic knot-like assembly (*Figure 1A*).

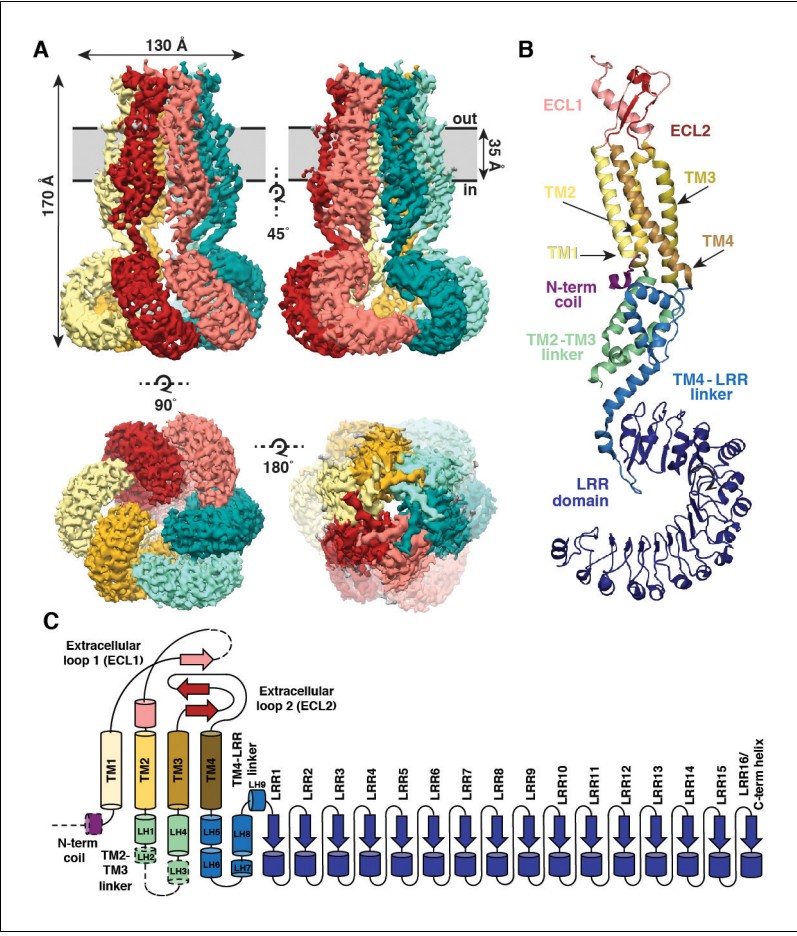

**Figure 1.** Overall architecture of homomeric SWELL1. (**A**) Cryo-EM reconstruction of SWELL1 homohexamer viewed from the membrane plane highlighting a dimer pair (top left, red and pink subunits) and an interface between dimers (top right, pink and green subunits), from the extracellular side (bottom left), and from the cytosolic side (bottom right). (**B**) Detailed view of SWELL1 'inner' protomer. (**C**) Topology diagram denoting secondary structural elements. Dashed lines indicate unresolved regions on both protomers in a dimer pair, while dashed shape borders indicate regions that are only resolved on one protomer.

DOI: https://doi.org/10.7554/eLife.38461.003

The following figure supplements are available for figure 1:

**Figure supplement 1.** SWELL1 overexpression in cells lacking other LRRC8 subunits produces small DCPIB-sensitive swelling-induced whole cell currents.

DOI: https://doi.org/10.7554/eLife.38461.004

**Figure supplement 2.** Purification of SWELL1-FLAG.

DOI: https://doi.org/10.7554/eLife.38461.005

**Figure supplement 3.** Cryo-EM data collection.

DOI: https://doi.org/10.7554/eLife.38461.006

**Figure supplement 4.** Model-to-map fit of electron density.

DOI: https://doi.org/10.7554/eLife.38461.007

**Figure supplement 5.** SWELL1 subunit pair and structural homology of SWELL1 structure to connexin-26 and innexin-6 structures.

DOI: https://doi.org/10.7554/eLife.38461.008

**Figure supplement 6.** Alignment of LRRC8 subunits (res 1–435).

DOI: https://doi.org/10.7554/eLife.38461.009

**Figure supplement 7.** Alignment of LRRC8 subunits (res 436–810).

DOI: https://doi.org/10.7554/eLife.38461.010

Perhaps the most striking architectural feature of VRAC is the symmetry mismatch between the cytosolic LRR domains and the pore-forming domains of the channel, despite its homo-hexameric assembly (*Figure 2*). The ECDs, TMs, and ILDs all share the same 6-fold symmetric arrangement (*Figure 2B*); however, in the cytosol, LRR domains dimerize in a parallel fashion with each LRR at either a 10 or −20° offset relative to the rest of its protomer, producing a 3-fold symmetric trimer of dimers (*Figure 2C*). The nonequivalence between identical subunits arises from a hinge around the conserved residue L402 in a helix of the TM4-LRR linker (*Figure 2D* and *Figure 1—Figure supplements 6* and *7*). This hinge allows the LRR domains to shift as rigid bodies, producing sufficient flexibility for them to interface at their edges via several charged residues (*Figures 2D* and *3A*). As a result, the helical C-termini of the two subunits in a dimer pair make two different sets of interactions with the neighboring LRR (*Figure 3B*). Focused 3D classification of the LRR domains revealed several arrangements of LRRs suggesting that flexibility of the LRR domains may play a functional role in channel gating (*Figure 2—figure supplement 1*), similar to the intracellular domains of the CorA magnesium channel (*Matthies et al., 2016*). Interestingly, the outer LRR subunit in the dimer exhibits helical density in the C-terminal half of the TM2-TM3 linker that rests on top of the outer proto-mer's LRR domain, adding an additional layer of intricacy to the network of cytosolic interactions (*Figure 1—figure supplement 5A–B*). Symmetry mismatch is also observed in the homotetrameric AMPA receptor GluA2, which similarly forms local dimers in different domain layers (*Sobolevsky et al., 2009*). Furthermore, the dimer-of-dimers topology of homotetrameric AMPA-subtype ionotropic glutamate receptors (iGluRs) defines the subunit organization of di- and tri-het-eromeric NMDA-subtype iGluR structures (*Karakas and Furukawa, 2014*; *Lee et al., 2014*;

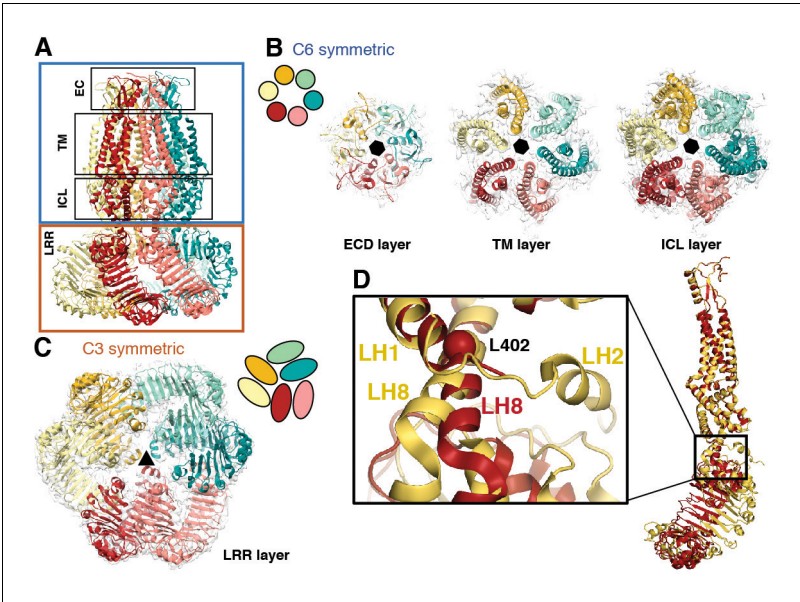

**Figure 2.** Subunit arrangement exhibits symmetry mismatch. (A) SWELL1 model viewed from the membrane plane with domain layers viewed perpendicular to the symmetry axis. (B–C) Domain layers viewed from the top of the channel grouped according to shared symmetry with simple schematic to demonstrate subunit arrangement. (B) From left to right: extracellular domain layer (EC), transmembrane domain layer (TM), and intracellular linker domain layer (ICL) all share the same 6-fold rotation symmetry axis (black hexagon). (C) The LRR domain layer has 3-fold rotational symmetry (black triangle), resulting from parallel pairing of three sets of LRR domains. (D) Asymmetry in LRR pairing arises from a hinge at L402 on LH8 that allows rotation of the LRR domain as a rigid body in a dimer pair. The first two TM domains of the inner (red) and outer (yellow) subunits are aligned to one another using the PyMOL align function.

DOI: https://doi.org/10.7554/eLife.38461.011

The following figure supplement is available for figure 2:

**Figure supplement 1.** Flexibility in LRR domains observed during 3D classification.
DOI: https://doi.org/10.7554/eLife.38461.012

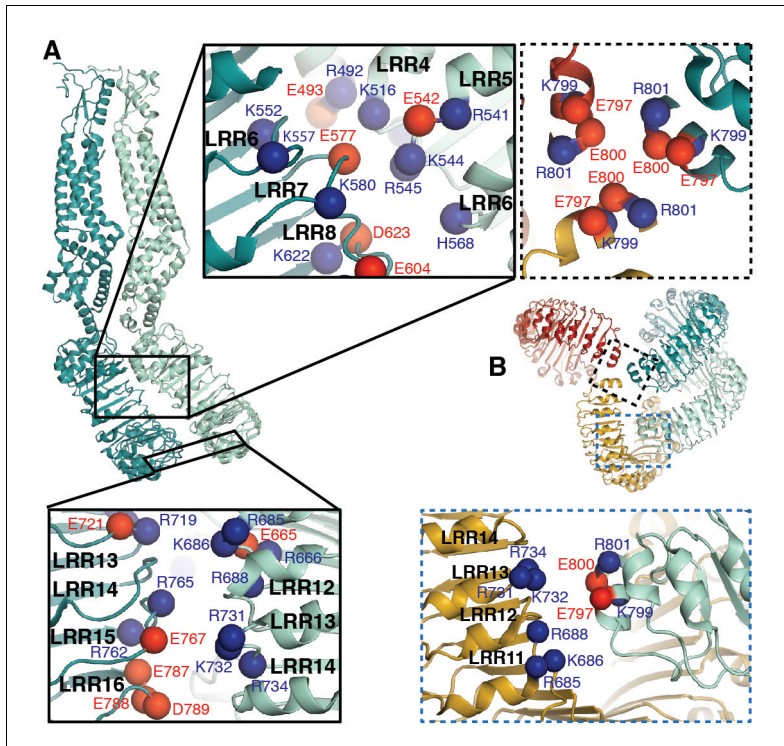

**Figure 3.** LRRs interact via charged residues at dimer interfaces and C-termini. (**A**) One dimer of SWELL1 subunits. Charged residues both of opposite and similar charges face each other in the interface between the two LRR domains (insets, top middle and bottom left; blue spheres are positively charged residues (Arg, Lys, and His), red spheres are negatively charged residues (Asp and Glu)). (**B**) C-termini of the two protomers in a dimer interact with regions of the neighboring LRR domain. Two of three 'outer' subunits are removed for clarity. 'Inner' subunits may be able to coordinate with one another via a triad of charged residues (E800) at their C-termini (inset, dashed border, top right), while the C-termini of the 'outer subunit' may interact with the edge of the neighboring outer subunit via charged residues R688 on LRR12 and/or K732 on LRR13 (inset, dashed blue border, bottom right).
DOI: https://doi.org/10.7554/eLife.38461.013

*Lü et al., 2017*). By analogy, we speculate that the trimer-of-dimers assembly of SWELL1 is recapitulated in, and influences the composition of, heteromeric VRACs.

Unlike other ion channels, there is little domain swapping between the subunits of the pore-forming domains of the SWELL1 channel. The individual helical bundles are loosely packed with one another and lined with hydrophobic residues. The inter-protomer space may be occupied by hydrophobic membrane components like lipid or cholesterol that might be important for channel assembly or lipid signaling. Such densities are observed in the inter-subunit space in innexin-6 and have been proposed to have a stabilizing role in the conformation of the helix bundles (*Oshima et al., 2016*). At the upper faces of the extracellular domains, on mostly flexible loops, resides a three residue KYD motif previously shown to be involved in voltage-dependent inactivation and selectivity (*Ullrich et al., 2016*); interestingly, KYD extends laterally towards the neighboring subunit (*Figure 4—figure supplement 1*), suggesting that subunit interactions in this region contribute to these channel properties.

The ECDs, TMs, and ILDs of all six subunits contribute to the ion-conducting pore (*Figure 4A–B*). Below that, windows of 35 by 40 Å between LRR dimer pairs are sufficiently large to allow ions and osmolytes to freely pass. In the extracellular domain, 25 Å above the membrane, a ring of arginines (R103) at the N-terminal tip of the extracellular helix forms the narrowest constriction in the channel structure (*Figure 4A–C*). We hypothesized that these arginines, only conserved between SWELL1 and the LRRC8B subunit (R99) (*Figure 1—figure supplement 6*), might directly interact with permeant anions. To test this hypothesis, we mutated positively-charged R103 to phenylalanine, and determined whether ion selectivity was altered in SWELL1-R103F + LRRC8C heteromeric channels

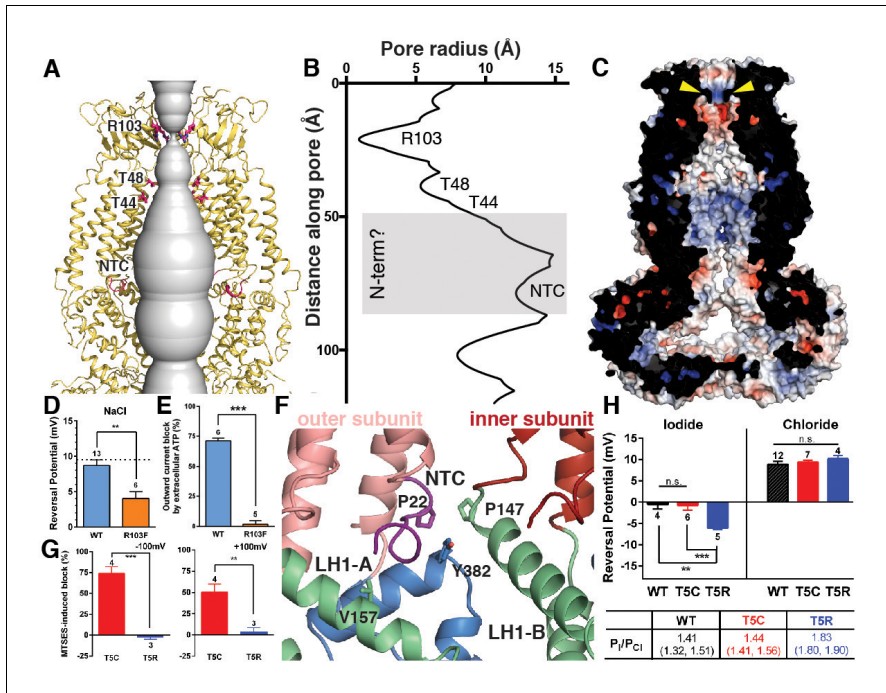

**Figure 4.** Ion pore structure and characterization of channels with mutations at pore-contributing residues R103 and T5. (A) Cartoon model of the SWELL1 pore, with two subunits removed for clarity. A surface representation of the radial distance between the protein surface and the pore axis is shown in grey. Pore-facing residues R103, T48 and T44, and N-terminal coil (NTC) are labeled in pink. (B) Graph of van der Waals radii of the pore, plotted against distance along the pore axis. Locations of residues R103, T48, T44, and NTC are labeled along 2D plot. Grey box covers potential area the N-terminus might occupy. (C) Electrostatic surface potential of channel pore, viewed by vertical cross-section. Narrow constriction on the extracellular side of the channel is formed by a ring of R103 residues (yellow arrows). Calculated using APBS implemented by Pymol2.0 with potentials ranging from −10 kT (red) to +10 kT (blue). (D–E) Cells expressing heteromeric VRACs composed of mutant SWELL1-R103F + LRRC8C show reduced chloride selectivity and insensitivity to external ATP block. (D) For highly Cl⁻ selective channels, the voltage at which there is no net current ($V_{rev}$) is close to the equilibrium potential for Cl⁻ (in these experiments $E_{Cl}$ = +9.75 mV; indicated by the dotted line). $V_{rev}$ of currents mediated by SWELL1-R103F-containing channels (orange bar; +4.6 ± 1.0 mV (mean ± s.e.m., n = 6 cells from 3 separate transfections)) is significantly reduced compared to WT (blue bar; +8.8 ± 0.8 mV (n = 13 from 6 separate transfections); p = 0.003, Student's t-test. (E) The percent block of whole cell leak subtracted hypotonic-induced currents by extracellular applied $Na_2ATP$ (2 mM) was determined at +100 mV. Outward WT SWELL1 + LRRC8C-mediated currents are blocked 72 ± 2% (mean ± s.e.m., n = 7 from 4 separate transfections; blue bar). Outward currents mediated by SWELL1-R103 + LRRC8C are not blocked by extracellular ATP (2 ± 3% (mean ± s.e.m., n = 5 from 3 separate transfections; orange bar); this difference is highly significant (p = 5.4e⁻8, Student's t-test). (F) Detailed view of coordination of NTC (purple). The NTC makes intrasubunit contacts with V157 on LH1 and a conserved Y382 at the kink between LH6 and LH7 of the TM4-LRR linker. Additionally, P22 of the NTC makes an intersubunit contact with a conserved P147 at the kink between TM2 and the TM2-TM3 linker of the neighboring subunit. (G–H) SWELL1-T5 is close to or part of the pore. (G) A cysteine mutation at SWELL1-T5 confers sensitivity to the polar MTS reagent MTSES applied extracellularly; maximum percent block of T5C-containing channels (red bars) by 3.33 mM MTSES was 74.2 ± 7.7% at −100 mV (left) and 50.4 ± 9.5% at +100 mV (right) (n = 5; mean ± s.e.m.; four separate transfections). The unmodifiable T5R-containing heteromeric channels (blue bars) are unaffected (n = 3 from 3 separate transfections; p=0.0009 at −100 mV and p=0.010 at +100 mV, Student's t-test). (H) Relative permeability $P_I/P_{Cl}$ is enhanced by the T5R mutation. Reversal potentials in iodide (left) and chloride (right) 230 mOsm/kg solutions are shown for the number of cells from 3 to 7 separate transfections (WT-, T5C- and T5R-expressing cells were from 6, 4, and 3 transfections, respectively, in the Cl⁻ condition, and 3, 3, and 3 transfections, respectively, in the I⁻ condition). The $V_{rev}$ of currents mediated by SWELL1-T5R + LRRC8C (blue) in I⁻ solution was significantly more negative than either WT- (black) or T5C (red)-containing channels (p=0.0088 (**) and 0.0047 (***), respectively. Table: $P_I/P_{Cl}$ is shown as means with lower and upper 95% confidence intervals.
DOI: https://doi.org/10.7554/eLife.38461.014

The following figure supplements are available for figure 4:

*Figure 4 continued on next page*

*Figure 4 continued*

**Figure supplement 1.** Additional characterization of selectivity at extracellular residues.
DOI: https://doi.org/10.7554/eLife.38461.015
**Figure supplement 2.** Representative data from cells expressing mutations in SWELL1 at T5 in the unresolved
N-terminal region in heteromeric VRAC.
DOI: https://doi.org/10.7554/eLife.38461.016

heterologously expressed in HeLa *LRRC8(A,B,C,D,E)^{-/-}* cells. We determined the reversal potential ($V_{rev}$) for hypotonicity-induced $Cl^-$ currents mediated by SWELL1-R103F + LRRC8C channels. The $V_{rev}$ of currents mediated by SWELL1-R103F + LRRC8C was significantly reduced compared to wildtype channels, indicating that the channels are less selective for $Cl^-$ (*Figure 4D*) (*Ackerman et al., 1994*; *Jackson and Strange, 1995*; *Tsumura et al., 1996*). Furthermore, extracellular ATP at concentrations that block ~75% of wildtype VRAC currents was ineffective on channels containing R103F (*Figure 4E* and *Figure 4—figure supplement 1*). Therefore, R103 is a critical residue within SWELL1 that impacts ion selectivity as well as pore block of heteromeric VRAC channels.

Within the pore, constrictions are observed at pore-facing residues T44 and T48 (*Figure 4A–B*). Interestingly, we had previously identified residue T44 via the substituted cysteine accessibility method (SCAM) on heteromeric channels as likely to be at or near the pore (*Qiu et al., 2014*). Near the bottom of the pore cavity, a constriction at the intracellular face of the membrane corresponds to a short N-terminal coil (NTC) sitting parallel to the inner leaflet of the membrane. The first 14 residues of the N-terminus of the channel are not resolved in the cryo-EM density, presumably due to flexibility. The absence of these residues is conspicuous; in the Cx26 and innexin-6 structures, an N-terminal helix forms a pore funnel structure that is the narrowest constriction in the structures of these channels and is thought to contribute to trafficking, selectivity, and gating (*Kyle et al., 2008*; *Maeda et al., 2009*; *Oshima, 2014*; *Oshima et al., 2016*). In our reconstruction, the short portion of the NTC that is resolved is highly coordinated by cytosolic domains and positioned to respond to conformational changes in the cytosolic domains of one protomer, as well as movements of the neighboring protomer (*Figure 4F*). Due to the similarities in pore structure between VRAC and connexin/innexin (*Figure 1—figure supplement 5*), we conducted functional assays to interrogate the role of the NTC in VRAC. We focused on residue T5 because the homologous residue is involved in stabilizing the pore funnel through a hydrogen bonding network in the Cx26 structure (*Maeda et al., 2009*). We made the mutation T5C to test whether extracellular addition of the negatively-charged, membrane-impermeable thiol-reactive reagent, 2-sulfonatoethyl methanethiosulfonate (MTSES), could alter VRAC activity in heteromeric channels composed of SWELL1-T5C + LRRC8C in HeLa *LRRC8(A,B,C,D,E)^{-/-}* cells via cysteine modification. While MTSES has no effect on wildtype heteromeric channels (*Qiu et al., 2014*) or channels containing SWELL1-T5R (*Figure 4G*), whole-cell currents mediated by SWELL1-T5C + LRRC8C are strongly suppressed upon the addition of MTSES, suggesting that T5C is part of a constriction narrow enough to block the pore upon covalent modification by MTSES (*Figure 4G* and *Figure 4—figure supplement 2*). We next determined the role of T5 in anion selectivity. Although SWELL1-T5C-containing channels have similar relative permeability to wildtype, SWELL1-T5R-containing channels are significantly more selective to iodide compared to chloride, confirming that this residue is close to or part of the channel pore (*Figure 4H* and *Figure 4—figure supplement 2*). Thus, the unresolved portion of the N-terminus plays a role in pore constriction in native channels composed of SWELL1 and LRRC8C. Its absence in our structure is likely due to either the high flexibility of the region or a peculiarity of the homomeric assembly of the channel.

## Discussion

Here we report the architecture and homo-hexameric assembly of SWELL1 channels. Electrophysiological analyses presented here demonstrate that the homomeric SWELL1 structure retains properties of more complex heteromers, as mutations based on the structure proved to be relevant for VRAC currents in a cellular context. The structure of SWELL1 also provides hints as to how VRAC gating is regulated. Since decreases in intracellular ionic strength cause activation (*Syeda et al., 2016*), gating would likely be initiated by movement of intracellular domains in response to changes

in salt concentration. We speculate that the multitude of charge-mediated interactions in the LRRs endows the SWELL1 structure with ionic-strength sensitivity, and via interactions with the N-terminus, the ILDs couple LRR movement to the transmembrane channel.

# Materials and methods

## Key resources table

| Reagent type (species) or resource | Designation | Source or reference | Identifiers | Additional information |
|---|---|---|---|---|
| Gene (*Homo sapiens*) | LRRC8A | Origene | Uniprot: Q8IWT6 | |
| Cell line (*Homo sapiens*) | Freestyle 293-F | ThermoFisher | RRID: CVCL_D603 | |
| Cell line (*Homo sapiens*) | HeLa | ATCC | RRID: CVCL_0030 | |
| Recombinant DNA reagent | pcDNA3.1/Zeo(-) | ThermoFisher | cat no: V86520 | |
| Chemical compound | digitonin | Sigma-Aldrich | CAS Number: 11024-24-1 | |
| Software, algorithm | MotionCor2 | doi:10.1038/nmeth.4193 | | |
| Software, algorithm | EMHP | doi:10.1093/bioinformatics/btx500 | | |
| Software, algorithm | Gctf | doi:10.1016/j.jsb.2015.11.003 | | |
| Software, algorithm | FindEM template correlator | doi:10.1016/j.jsb.2003.11.007 | | |
| Software, algorithm | cryoSPARC | doi: 10.1038/nmeth.4169 | | https://cryosparc.com/ |
| Software, algorithm | RELION | | RRID:SCR_016274 | |
| Software, algorithm | Rosetta | | RRID:SCR_015701 | https://www.rosettacommons.org/software |
| Software, algorithm | Robetta | | | http://robetta.bakerlab.org/ |
| Software, algorithm | COOT | | RRID:SCR_014222 | http://www2.mrc-lmb.cam.ac.uk/personal/pemsley/coot/ |
| Software, algorithm | Phenix | | RRID:SCR_014224 | https://www.phenix-online.org/ |
| Software, algorithm | PyMOL | PyMOL Molecular Graphics System, Schrodinger LLC | RRID:SCR_000305 | http://www.pymol.org/ |
| Software, algorithm | UCSF Chimera | UCSF | RRID:SCR_004097 | http://plato.cgl.ucsf.edu/chimera/ |

## CRISPR LRRC8 KO cell lines

Knock-out of LRRC8 genes in HeLa and suspension Freestyle HEK293-F cell line was completed using CRISPR/Cas9-mediated gene disruption (*Ran et al., 2013*). *SWELL1 (LRRC8A), LRRC8B, LRRC8D*, and *LRRC8E* genes were targeted using guideRNA (gRNA) sequences reported by *Voss et al. (2014)*; the LRRC8C gene was targeted with a gRNA sequence reported by *Syeda et al. (2016)*. Cloning of the gRNAs into PX458-mCherry plasmid was completed as reported in *Syeda et al. (2016)*. Multiple plasmids were transfected simultaneously using either Lipofectamine 2000 or PEI max. After 48–72 hr, fluorescent mCherry positive cells were single-cell sorted into 96-well plates. Successful knock-out was determined by genotyping targeted regions for frameshift mutations and verified by mass spectrometry analysis. For HeLa cells (*LRRC8*$^{-/-}$ HeLa cells), complete knock-out was verified for all five LRRC8 genes. For HEK293-F suspension cells, complete knock-out was verified for *LRRC8B-E* (*LRRC8(B,C,D,E)*$^{-/-}$ HEK293-F cells). One *SWELL1* allele remained intact in all surviving suspension culture lines. All cell lines tested negative for mycoplasma contamination.

## Protein expression and purification

Human *SWELL1 (LRRC8A)* (Origene #RC208632) was cloned with a C-terminal FLAG-tag (DYKDDDDK) separated by a triple glycine linker (*SWELL1-GGG-FLAG*) into a pcDNA3.1/Zeo(-) vector using Gibson cloning. HEK293-F *LRRC8(B,C,D,E)-/-* cells were transfected at a cell density of $1.8*10^6$ cells/mL with 1 mg/L cells of *SWELL1-GGG-FLAG* plasmid DNA combined with 3 mg/L cells

of PEI max. After 48 hr, cells were pelleted and solubilized in solubilization buffer (20 mM Tris pH 8, 150 mM NaCl, 1% DMNG, 2 mg/mL iodoacetamide, and EDTA-free protease inhibitor cocktail (PIC)) at 4°C with vigorous shaking. The cell lysate was ultracentrifuged at 90,000 $x$ $g$ for 30 min at 4°C and the supernatant was collected and combined with 1 mL/L cells of FLAG M2 affinity resin for 1 hr batch incubation at 4°C with gentle shaking. Resin was washed in a gravity column with 5 mL per mL of resin (column volumes; CV) of solubilization buffer (20 mM Tris pH 8, 150 mM NaCl, 1% DMNG, 2 mg/mL iodoacetamide, and EDTA-free PIC), 5CV of high salt wash buffer (20 mM Tris pH 8, 150 mM NaCl, 0.05% digitonin, and EDTA-free PIC), and 10CV of wash buffer (20 mM Tris pH 8, 150 mM NaCl, 0.05% digitonin, and EDTA-free PIC). Protein was eluted using elution buffer (20 mM Tris pH 8, 150 mM NaCl, 0.05% digitonin, EDTA-free PIC and 3x FLAG peptide (Sigma or in-house peptide production)). Sample was concentrated and injected onto Shimadzu HPLC and separated using a Superose 6 Increase column equilibrated with running buffer (20 mM Tris pH 8, 150 mM NaCl, 0.05% digitonin, and EDTA-free PIC). The peak corresponding to SWELL1 homomeric oligomers (~800 kDa) was collected and used for cryo-EM grid preparation. The sample was concentrated to ~8 mg/mL using 100 kDa MWCO concentrators. Protein (3 µl) was applied to plasma cleaned UltrAuFoil 1.2/1.3 300 mesh grids, blotted for 6 s with 0 blot force, and plunge frozen into nitrogen cooled liquid ethane using a Vitrobot Mark IV (ThermoFisher).

## Cryo-EM data collection

Images were collected at 200 kV on a Talos Arctica electron microscope (ThermoFisher) with a K2 direct electron detector (Gatan) at a nominal pixel size of 1.15 Å. Leginon software was used to automatically collect micrographs (*Suloway et al., 2005*). The total accumulated dose was ~55 e⁻/Å² and the defocus range was 0.8–1.5 µm. Movies were aligned and dose-weighted using MotionCor2 (*Zheng et al., 2017*).

## Image processing

Images were assessed for quality and edges of gold holes were masked using EMHP (*Berndsen et al., 2017*). CTF values were estimated using Gctf (*Zhang, 2016*). Template-based particle picking was completed using FindEM template correlator (*Roseman, 2004*). Particles were extracted using Relion 2.1 (*Scheres, 2012*) then subjected to 2D classification using cryoSPARC (*Punjani et al., 2017*). 130,054 particles corresponding to good 2D class averages were selected for further data processing. An ab initio initial model was created in cryoSPARC followed by iterative angular reconstitution and reconstruction. The resulting density map was used as a seed for refinement of the data set in Relion 2.1. Resolution of the resulting map was 4.6 Å. The map showed significant disorder in the LRR regions; however the map reveals that LRR regions arrange pairwise around a three-fold symmetry axis. As the transmembrane and extracellular domains were well-resolved, refinement was pursued imposing C3 symmetry and introducing a mask that excluded density outside of the well-defined, three-fold symmetric transmembrane/extracellular domains. Resolution of the resulting map was 4.0 Å; transmembrane/extracellular domains were well-resolved whereas LRR regions were largely disordered. This map was then used to create suitable projections that were subtracted from particles, thereby creating a particle data set corresponding mostly to LRR densities. This new data set was then subjected to 3D classification in Relion 2.1 (K-means split of 12). One of the resulting classes showed order in the pairwise LRR arrangement around the three-fold symmetry axis. Particles corresponding to this class (25,719) were then refined locally around the previously obtained coordinate assignment imposing three-fold symmetry resulting in an LRR density map at 5.0 Å resolution. Additionally – due to the overall higher degree of order – original particles corresponding to the 25,719 density-subtracted particles were refined under three-fold symmetry constraints. Resolution of the resulting map was 4.4 Å.

## Model building and refinement

An initial model of an N-terminal portion of SWELL1 was generated with RobettaCM using innexin-6 (5H1Q) as a template structure (*Oshima et al., 2016*; *Song et al., 2013*). The SWELL1 topology was predicted using OCTOPUS (*Viklund and Elofsson, 2008*). Predicted transmembrane regions were manually aligned to the transmembrane helices of the template structure 5H1Q (*Oshima et al., 2016*). Intervening regions of SWELL1 were aligned to 5H1Q using BLASTp. 10,000

independent homology models were generated with RosettaCM and clustered using Calibur (*Li and Ng, 2010*). The resulting model with the lowest Rosetta energy from the largest cluster was used as a guide for *ab initio* building of the transmembrane helices, extracellular domains, and intracellular linker domain. Sequence register was aided by bulky side chains and disulfide bonds in the extracellular domain. A Robetta-generated model of the SWELL1 LRR domain was docked into the EM density corresponding to the LRR of the outer subunit, which was better resolved than the inner subunit (*Kim et al., 2004*). This LRR model was adjusted manually to fit the density, then copied and docked into the LRR density of the inner subunit, followed by further adjustments. During the building process, manual building in COOT (*Emsley and Cowtan, 2004*) was iterated with real space refinement using Phenix (*Adams et al., 2010*) or RosettaRelax (*DiMaio et al., 2009*). Structures were evaluated using EMRinger (*Barad et al., 2015*) and MolProbity (*Chen et al., 2010*). The final model contains residues 15–68, 94–174, 232–802 in the inner subunit and 15–68, 94–175, 214–802 in the outer subunit. Side chains of residues 15–21, 359–364, 787–802 of both subunits and 214–233 of the outer subunit were trimmed to Cβ because of limited resolution and lack of well-defined secondary structures in these regions. Structure figures were made in Pymol (*Schrodinger,, 2017*) and UCSF Chimera (*Pettersen et al., 2004*). Pore radii were calculated using HOLE (*Smart et al., 1996*). The APBS plugin in pymol was used to calculate surface representations of electrostatic potentials.

## Electrophysiology and cell culture

Electrophysiology experiments were completed with HeLa *LRRC8*$^{-/-}$ cells. HeLa *LRRC8*$^{-/-}$ cells were transfected 1–3 days earlier with SWELL1 constructs together with LRRC8C-ires-GFP in a 2:1 ratio (0.8 and 0.4 γ/ml for each coverslip). VRAC currents using a 2:1 ratio of SWELL1:LRRC8C were at least twice as large as those using a 1:1 ratio (data not shown). Only one cell per coverslip was tested for its response to hypotonic solution. In experiments aimed at determining whether HeLa *LRRC8*$^{-/-}$ cells transfected with SWELL1 only could express VRAC currents, the extracellular solution contained (in mM) 90 NaCl, 2 KCl, 1 $MgCl_2$, 1 $CaCl_2$, 10 HEPES, 110 mannitol (isotonic, 300 mOsm/kg) or 30 mannitol (hypotonic, 230mOsm/kg), pH 7.4 with NaOH; recording pipettes were filled with intracellular solution containing (in mM): 133 CsCl, 5 EGTA, 2 $CaCl_2$, 1 $MgCl_2$, 10 HEPES, 4 Mg-ATP, 0.5 Na-GTP (pH 7.3 with CsOH; 106 nM free $Ca^{2+}$) and had resistances of 2–3 MΩ. Experiments testing R103F and T5 mutants used extracellular solutions described in *Qiu et al. (2014)* ('bianionic') and intracellular solution used in *Syeda et al. (2016)* (130 mM CsCl, 10 HEPES, 4 Mg-ATP, pH 7.3). These were used to determine relative permeability $P_I/P_{Cl}$. An agar bridge was used between the ground electrode and the bath in all experiments.

## Acknowledgements

We thank H Turner, W Anderson, C Bowman, T Nieusma, R Hoffman, and L Kubalek for training in electron microscopy and computational methods. We acknowledge A Coombs for molecular biology assistance. We acknowledge P Dawson and P Cistrone for assistance in the production of 3x FLAG peptide reagent, and B Seegers for assistance with FACS for the development of CRIPSR cell lines. We thank R MacKinnon, G Lander, S Murthy and members of the Ward and Patapoutian labs for helpful discussions. This work was supported by National Institutes of Health (NIH) National Research Service Award F31 NS093778 to JMK, a Ray Thomas Edwards Foundation grant to ABW, and NIH grant NS083174 to AP. AP is an investigator of Howard Hughes Medical Institute (HHMI).

## Additional information

### Competing interests

Stuart M Cahalan: Currently affiliated with Vertex Pharmaceuticals. The other authors declare that no competing interests exist.

## Funding

| Funder | Grant reference number | Author |
| --- | --- | --- |
| National Institutes of Health | F31 NS093778 | Jennifer M Kefauver |
| Ray Thomas Edwards Foundation | | Andrew B Ward |
| National Institutes of Health | NS083174 | Ardem Patapoutian |
| Howard Hughes Medical Institute | | Ardem Patapoutian |

The funders had no role in study design, data collection and interpretation, or the decision to submit the work for publication.

## Author contributions

Jennifer M Kefauver, Conceptualization, Data curation, Formal analysis, Funding acquisition, Validation, Investigation, Visualization, Methodology, Writing—original draft, Writing—review and editing, Developed sample preparation protocols, Expressed and purified protein, Froze cryo-EM samples, Collected and processed cryoEM data, Built and refined atomic models; Kei Saotome, Formal analysis, Validation, Visualization, Writing—original draft, Writing—review and editing, Built and refined atomic models; Adrienne E Dubin, Conceptualization, Data curation, Formal analysis, Validation, Investigation, Visualization, Methodology, Writing—original draft, Writing—review and editing, Designed and performed whole cell electrophysiology experiments; Jesper Pallesen, Formal analysis, Validation, Visualization, Methodology, Writing—review and editing, Processed cryoEM data; Christopher A Cottrell, Formal analysis, Validation, Writing—review and editing, Refined atomic models; Stuart M Cahalan, Resources, Methodology, Writing—review and editing, Generated CRISPR cell lines; Zhaozhu Qiu, Resources, Methodology, Writing—review and editing, Generated CRIPSR cells lines and developed sample preparation protocols; Gunhee Hong, Investigation, Methodology, Writing—review and editing, Developed sample preparation protocols and created mutant constructs; Christopher S Crowley, Methodology, Writing—review and editing, Developed sample preparation protocols; Tess Whitwam, Wen-Hsin Lee, Investigation, Writing—review and editing, Created mutant constructs; Andrew B Ward, Ardem Patapoutian, Conceptualization, Resources, Supervision, Writing—original draft, Project administration, Writing—review and editing

## Author ORCIDs

Jennifer M Kefauver (iD) http://orcid.org/0000-0002-6818-4673
Jesper Pallesen (iD) https://orcid.org/0000-0002-3270-1587
Ardem Patapoutian (iD) https://orcid.org/0000-0003-0726-7034

## Decision letter and Author response

Decision letter https://doi.org/10.7554/eLife.38461.025
Author response https://doi.org/10.7554/eLife.38461.026

# Additional files

## Supplementary files

• Supplementary file 1. Cryo-EM data collection, refinement and validation statistics.
DOI: https://doi.org/10.7554/eLife.38461.017
• Transparent reporting form
DOI: https://doi.org/10.7554/eLife.38461.018
• Reporting standard 1
DOI: https://doi.org/10.7554/eLife.38461.019

## Data availability

The cryo-EM map of human SWELL1 was deposited into the Electron Microscopy Data Bank with accession code 7935. The atomic model of human SWELL1 was deposited into the Protein Data Bank with PDB ID 6DJB.

The following datasets were generated:

| Author(s) | Year | Dataset title | Dataset URL | Database, license, and accessibility information |
|---|---|---|---|---|
| Kefauver JM, Pallesen J, Kei Saotome, Christopher A Cottrell, Andrew B Ward, Ardem Patapoutian | 2018 | Structure of the human volume regulated anion channel | https://www.ebi.ac.uk/pdbe/entry/emdb/EMD-7935 | Publicly available at Electron Microscopy Data Bank (accession no: EMD-7935) |
| Kefauver JM, Saotome K, Pallesen J, Cottrell CA, Ward AB, Patapoutian A | 2018 | Structure of the human volume regulated anion channel | https://www.rcsb.org/structure/6DJB | Publicly available at RCSB Protein Data Bank (accession no: 6DJB) |

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
