## [Decision Letter]

Thank you for submitting your article "Structure of the Human Volume Regulated Anion Channel" for consideration by *eLife*. Your article has been reviewed by three peer reviewers, and the evaluation has been overseen by Kenton Swartz as the Reviewing Editor and Richard Aldrich as the Senior Editor. The following individual involved in review of your submission has agreed to reveal his identity: Joseph A Mindell (Reviewer #3).

The reviewers have discussed the reviews with one another and the Reviewing Editor has drafted this decision to help you prepare a revised submission.

Summary:

This manuscript presents a cryo-EM-derived structure of a SWELL1 ion channel, which provides key information about the SWELL, or LRRC8, volume-regulated anion channels. The structure presented here is of a SWELL1 (LRRC8A) homohexamer; though this is not the biological form of the channel, which needs contributions from other subunits, this protein does form functional volume-regulated channels in both lipid bilayers (previously published) and in HeLa cells lacking the native VRAC. The hexameric structure presented here is noted to be related to the known connexins and innexins, which also form hexameric, large-pore channels. A notable feature of the assembly is a symmetry mismatch between the transmembrane and cytoplasmic domains – where the TMs are in a sixfold arrangement, the cytoplasmic domains form a trimer of dimers. A notable feature in the structure is a narrowing due to a loop on the extracellular face. Mutations to the arginine residue at this location diminish the anion selectivity of the channel, consistent with a role in ion selectivity. The work is generally clearly presented and represents an important early point in the understanding of this fairly new family of ion channels. The quality of the data is generally high, but there are few issues regarding reproducibility and interpretation that must be addressed.

Essential revisions:

1) Have you tried or considered using symmetry expansion as described in these studies?

N. Grigorieff, Frealign: An exploratory tool for single-particle cryo-EM. Methods Enzymol. (2016)

M. Zhou, Y. Li, Q. Hu, X. C. Bai, W. Huang, C. Yan, S. H. W. Scheres, Y. Shi, Atomic structure of the apoptosome: Mechanism of cytochrome c- and dATP-mediated activation of Apaf-1. Genes Dev (2015)

H. E. Autzen, A. G. Myasnikov, M. G. Campbell, D. Asarnow, D. Julius, Y. Cheng, Structure of the human TRPM4 ion channel in a lipid nanodisc. Science 2017

2) The resolution map in Figure 1E shows resolution in the 4-5 Å range in the LRR domain (with the periphery becoming slightly worse). However, the LRR regions shown in Figure 1—figure supplement 4 appear to be less well resolved than suggested by the resolution map. Given that analysis of the LRR interfaces (a major part of the study) relies on this part of the map, additional metrics of map vs. model correlation would bolster reader confidence in the model analysis.

3) The presumed lipid, cholesterol or detergent densities should be shown in the context of the cryo-EM map (i.e. not solely with the molecular model as in Figure 3—figure supplement 1C). This will help the reader to assess the strength of these densities.

4) We disagree with the authors on their interpretation of the mutation at the TM subunit interface (main text, fifth paragraph, Figure 3—figure supplement 1). First, the mutation does not produce a "constitutively activated" channel. Rather it seems to introduce some kind of leak in the closed state; the data shown in Figure 3—figure supplement 1B for the mutant show robust activation by hypo-osmolality, albeit somewhat reduced from the wild type. There is not enough here to make a compelling argument about the role of Y127 in stabilizing the channel. In addition, the inhibition by DCPIB shown in panel C of that figure is described as having been repeated in one other cell. In general, the statistics and reproducibility information presented in the paper are excellent. This is an exception – even this control should at least be repeated in biological replicate to confirm the result. Given the lack of insight provided by the 127C mutation in general, the paper would be well served to have this figure simply removed.

5) Similarly, the T5 electrophysiology data in Figure 4—figure supplement 2 lacks any information about reproducibility or statistics. These must be added if this supplement will be included in the final paper.

---

## [Author Response]

Essential revisions:1) Have you tried or considered using symmetry expansion as described in these studies?N. Grigorieff, Frealign: An exploratory tool for single-particle cryo-EM. Methods Enzymol. (2016)M. Zhou, Y. Li, Q. Hu, X. C. Bai, W. Huang, C. Yan, S. H. W. Scheres, Y. Shi, Atomic structure of the apoptosome: Mechanism of cytochrome c- and dATP-mediated activation of Apaf-1. Genes Dev (2015)H. E. Autzen, A. G. Myasnikov, M. G. Campbell, D. Asarnow, D. Julius, Y. Cheng, Structure of the human TRPM4 ion channel in a lipid nanodisc. Science 2017

We thank the reviewers for their suggestion. Using the methods in Zhou, et al., (2015) as a model, we attempted symmetry expansion on the LRR dimer alone. Unfortunately, this did not improve resolution in this region, which is consistent with a certain level of continuous heterogeneity that prevented convergence at high resolution.

2) The resolution map in Figure 1E shows resolution in the 4-5 Å range in the LRR domain (with the periphery becoming slightly worse). However, the LRR regions shown in Figure 1—figure supplement 4 appear to be less well resolved than suggested by the resolution map. Given that analysis of the LRR interfaces (a major part of the study) relies on this part of the map, additional metrics of map vs. model correlation would bolster reader confidence in the model analysis.

We acknowledge the reviewers’ concern that the resolution in the LRR domains is limited due to flexibility. To increase confidence that the quality of the map in this region is resolved to ~5 Å, we include an additional panel in Figure 1—figure supplement 4 demonstrating beta-strand separation between LRR4-8. We agree that the resolution at the interfaces of the LRRs is not sufficient to build side chain densities, but we argue that the canonical fold of the LRR domains with hydrophobic residues internal to the solenoid give us confidence that the residues assigned to the loops between the alpha helix and beta strand of each repeat are accurate to the extent of the data. As such, all side chains in these regions have been trimmed to carbon beta atoms to prevent over-interpretation of the model. Additionally, we report separate EMRinger scores and map-model cross correlation for the entire model (Swell1), the pore-forming domains, and the LRR domains in Supplementary file 1.

3) The presumed lipid, cholesterol or detergent densities should be shown in the context of the cryo-EM map (i.e. not solely with the molecular model as in Figure 3—figure supplement 1C). This will help the reader to assess the strength of these densities.

This figure supplement has been removed in the resubmitted text. Please see response to essential revision #4.

4) We disagree with the authors on their interpretation of the mutation at the TM subunit interface (main text, fifth paragraph, Figure 3—figure supplement 1). First, the mutation does not produce a "constitutively activated" channel. Rather it seems to introduce some kind of leak in the closed state; the data shown in Figure 3—figure supplement 1B for the mutant show robust activation by hypo-osmolality, albeit somewhat reduced from the wild type. There is not enough here to make a compelling argument about the role of Y127 in stabilizing the channel. In addition, the inhibition by DCPIB shown in panel C of that figure is described as having been repeated in one other cell. In general, the statistics and reproducibility information presented in the paper are excellent. This is an exception – even this control should at least be repeated in biological replicate to confirm the result. Given the lack of insight provided by the 127C mutation in general, the paper would be well served to have this figure simply removed.

We agree with the reviewers that Y127C might not have constitutive activity. Our intention was to convey that currents mediated by SWELL1-Y127C subunit together with endogenous non-LRRC8A subunits are observed as soon as whole cell recordings are initiated (HeLa shA KD cells expressing shRNA against LRRC8A (Qiu et al., 2014)). We think it is likely that these channels are more easily activated under our experimental conditions than wildtype channels, but our current dataset does not allow conclusions about mechanism. Since we do not at this time have a compelling argument to state anything mechanistic about the role of this mutation, we have removed this supplemental figure and associated text from the resubmitted manuscript.

5) Similarly, the T5 electrophysiology data in Figure 4—figure supplement 2 lacks any information about reproducibility or statistics. These must be added if this supplement will be included in the final paper.

Figure 4—figure supplement 2 contains representative traces to support the data shown in Figure 4G and H. The title of Figure 4—figure supplement 2 was changed to “Representative data from cells expressing mutations in SWELL1 at T5 in the unresolved N-terminal region in heteromeric VRAC” to clarify that these panels contain example traces. Also, additional experiments were added to panels G and H in Figure 4 allowing statistical significance for each condition to be demonstrated.